# Ceramide Phosphoethanolamine as a Possible Marker of Periodontal Disease

**DOI:** 10.3390/membranes12070655

**Published:** 2022-06-25

**Authors:** Maja Grundner, Haris Munjaković, Tilen Tori, Kristina Sepčić, Rok Gašperšič, Čedomir Oblak, Katja Seme, Graziano Guella, Francesco Trenti, Matej Skočaj

**Affiliations:** 1Department of Biology, Biotechnical Faculty, University of Ljubljana, Večna pot 111, 1000 Ljubljana, Slovenia; maja.grundner@bf.uni-lj.si (M.G.); munjakovic.haris@gmail.com (H.M.); tilen.tori@gmail.com (T.T.); kristina.sepcic@bf.uni-lj.si (K.S.); 2Faculty of Medicine, University of Ljubljana, Vrazov trg 2, 1000 Ljubljana, Slovenia; rok.gaspersic@mf.uni-lj.si (R.G.); cedomir.oblak@mf.uni-lj.si (Č.O.); katja.seme@mf.uni-lj.si (K.S.); 3Bioorganic Chemistry Laboratory, Department of Physics, University of Trento, Via Sommarive 14, 38123 Trento, Italy; graziano.guella@unitn.it (G.G.); f.trenti@unitn.it (F.T.)

**Keywords:** ceramide phosphoethanolamine, periodontal disease, biomarker, aegerolysin, erylysin A, *Porphyromonas gingivalis*

## Abstract

Periodontal disease is a chronic oral inflammatory disorder initiated by pathobiontic bacteria found in dental plaques—complex biofilms on the tooth surface. The disease begins as an acute local inflammation of the gingival tissue (gingivitis) and can progress to periodontitis, which eventually leads to the formation of periodontal pockets and ultimately results in tooth loss. The main problem in periodontology is that the diagnosis is based on the assessment of the already obvious tissue damage. Therefore, it is necessary to improve the current diagnostics used to assess periodontal disease. Using lipidomic analyses, we show that both crucial periodontal pathogens, *Porphyromonas gingivalis* and *Tannerella forsythia*, synthesize ceramide phosphoethanolamine (CPE) species, membrane sphingolipids not typically found in vertebrates. Previously, it was shown that this particular lipid can be specifically detected by an aegerolysin protein, erylysin A (EryA). Here, we show that EryA can specifically bind to CPE species from the total lipid extract from *P. gingivalis*. Furthermore, using a fluorescently labelled EryA-mCherry, we were able to detect CPE species in clinical samples of dental plaque from periodontal patients. These results demonstrate the potential of specific periodontal pathogen-derived lipids as biomarkers for periodontal disease and other chronic inflammatory diseases.

## 1. Introduction

Periodontal disease is of great importance worldwide, affecting more than 10% of the adult population [1,2,3] and also occurring in children and adolescents [4]. It is an inflammatory event caused by dysbiosis of the commensal subgingival microbiota, which promotes the growth of pathobiontic bacteria, immune system activation, inflammation, and tissue breakdown [5,6,7]. More than 600 bacterial species are found in the human oral cavity, but only a few play a role in oral health [6]. A dental biofilm consists of aerobic bacteria that form microcolonies surrounded by a protective glycoprotein and polysaccharide matrix in healthy individuals. With good oral hygiene, the bacterial dental biofilm, plaque, can be removed, but otherwise, plaque begins to accumulate and cause gingival inflammation [7,8]. With inflammation, the microbiota changes to a predominance of Gram-negative and strictly anaerobic species [9], including *Porphyromonas gingivalis*, *Tannerella forsythia*, *Aggregatibacter actinomycetemcomitans*, *Treponema denticola*, *Fusobacterium nucleatum*, *Prevotella intermedia*, and some others [6,10].

The standard diagnostic procedure in periodontology is usually based on clinical measurements, such as probing pocket depth, clinical attachment level, and bleeding on probing, supported by radiographic evaluation [6]. As patients usually do not experience pain and do not seek medical attention, the disease can often reach an irreversible stage before treatment is initiated and cannot be controlled by oral hygiene [11]. Therefore, the search for diagnostic biomarkers is crucial for early diagnosis and has been the focus of research in the last 20 years [12,13,14,15,16,17,18]. More than 900 compounds have been found in gingival crevicular fluid [19], including inflammatory mediators, oxidative stress markers, degradation products, host-derived enzymes, growth factors, and mediators of bone homeostasis [5]. Saliva analysis is the alternative because it is easily accessible and a larger number of samples can be collected without requiring additional skills. Moreover, saliva reflects the pooled activity of all periodontal sites [18,20]. One of the currently available and promising biomarkers in saliva is neutrophil collagenase or matrix metalloproteinase 8 (MMP-8) [21,22], which belongs to a group of collagenases and is considered the most important collagenase in periodontitis [23,24]. Nevertheless, MMP-8 analyses should be used with caution, as some studies show higher MMP-8 levels in healthy individuals, while others show the same results in patients with periodontitis [25,26,27,28]. Furthermore, proteomic analysis, which is still in progress, could improve the diagnosis and treatment of periodontal disease [29]. However, there is no robust biochemical laboratory test routinely used to diagnose and monitor the progression of periodontal disease [5,30].

Ceramide phosphoethanolamine (CPE) is a major sphingolipid in the cell membranes of arthropods [31,32,33] and is also found in marine animals [34,35], some invertebrates [36,37,38], protozoa [39,40] and oomycetes [41]. Trace amounts of CPE can also be detected in mammalian cells (0.002–0.005% of all phospholipids), with the highest levels found in the testis and brain [42,43]. Lipidomic analyses have already shown the presence of CPE as well as its dihydrogenated form, dihydroceramide phosphoethanolamine (DH-CPE), in bacteria of the phylum Bacteroidetes [44,45] and also in dental plaque of patients with periodontal disease [46].

Since CPE is present in periodontal pathogens it could represent a potential molecule for the diagnosis of periodontal disease and the determination of its severity. The question of how to detect CPE can be addressed using a protein from the aegerolysin family of edible oyster mushrooms (*Pleurotus* sp.). Aegerolysins (Pfam 06355; InterPro IPR009413) are small (13–20 kDa) acidic proteins that have been identified in several eukaryotes and prokaryotes and are particularly abundant in bacteria and mushrooms [47,48]. The most prominent feature of all aegerolysins described to date is their ability to interact with specific membrane lipids and lipid domains [48,49,50]. Most of the currently characterized aegerolysins from oyster mushrooms can recognize and bind (*k_D_*~1 nM) to sphingomyelin (SM)/cholesterol (Chol)-enriched membranes [48,49] and even 1000-fold more strongly (*k_D_*~1 nM) to membranes containing equimolar amounts of CPE and Chol [51,52].

Aegerolysin erylysin A (Ery)A is a 15 kDa protein produced by *Pleurotus eryngii.* It is the only *Pleurotus* aegerolysin characterised to date that interacts exclusively with CPE-containing membranes and not with membranes containing SM in complex with Chol [51,53]. It can also bind efficiently to artificial lipid vesicles and biological membranes containing physiologically relevant concentrations (1–5 mol%) of CPE [49,54]. Since EryA binds exclusively to CPE without interfering with SM, which is the major sphingolipid of the mammalian membrane, it could be used for diagnostic applications in medicine. For example, fluorescently labelled EryA has already been used to detect CPE in the exoplasmic leaflet of the bloodstream form of *Trypanosoma brucei*. Since this parasite causes a severe disease, African sleeping sickness, EryA seems to be a suitable tool for visualizing the parasite in mammalian tissues [51]. In addition, EryA-mCherry also successfully labelled CPE-containing Sf9 insect cells [53].

Moreover, *Pleurotus* aegerolysins can act in concert with proteins that have a membrane-attack-complex/perforin (MACPF) domain and are also produced by *Pleurotus* mushrooms [55,56,57]. After binding to the membrane lipid receptor, the aegerolysins recruit the MACPF protein partner, which undergoes extensive conformational changes, penetrates the membrane, and causes cell lysis [56,57,58]. It has been previously shown that the combination of EryA and the MACPF partner pleurotolysin B (PlyB) forms transmembrane pores in artificial lipid membranes composed of CPE and is also cytotoxic to CPE-containing Sf9 insect cell line [53,58]. Even more, the cytotoxic activity was demonstrated against some invasive alien pests, such as western corn rootworm and Colorado potato beetle, which also contain CPE in their cell membranes [53].

Based on its specific interactions with CPE, we assumed that EryA might bind to CPE and its forms, such as DH-CPE, produced by periodontal bacteria. Therefore, the aim of this study was to test whether fluorescently labelled EryA could be used as a diagnostic tool to detect CPE species and for monitoring periodontal disease.

## 2. Materials and Methods

### 2.1. Chemicals

Unless specified otherwise, all chemicals used in the study were from Sigma-Aldrich (Boston, MA, USA). Cholesterol, 1-palmitoyl-2-oleoyl-*sn*-glycero-3-phosphocholine (POPC) and porcine brain SM were from Avanti Polar Lipids (Montgomery, AL, USA) and CPE from Matreya (Harrisburg, PA, USA). Lipids were dissolved in chloroform or in the case of CPE, in chloroform/methanol (9/1, *v*/*v*) and stored at −20 °C prior to use.

### 2.2. Bacteria

*Porphyromonas gingivalis* ATCC 33277, *Tannerella forsythia* DSM 102835, and *Aggregatibacter actinomycetecomitans* DO15 were provided by prof. Katja Seme from the Institute of microbiology and immunology, Faculty of Medicine, University of Ljubljana, Slovenia. As negative control *Escherichia coli* DH5α was used. All bacteria were maintained for 3–5 days at 37 °C in different growth conditions: *P. gingivalis* on 5% horse blood agar plates supplemented with 5 mg/L hemin and 1 mg/L menadione in an anaerobic atmosphere; *T. forsythia* on a tryptic soy agar containing 0.001% N-acetylmuramic acid in an anaerobic atmosphere; *A. actinomycetecomitans* on a tryptic soy-serum-bacitracin-vancomycin agar in air +5% CO_2_; and *E. coli* on Luria-Bertani agar in an aerobic atmosphere prior to use.

### 2.3. Proteins

Recombinant proteins EryA, fluorescently-tagged EryA-mCherry and PlyB were produced as described previously [53,56]. Protein sizes and purity were determined using SDS-PAGE (Bio-Rad, Sacramento, CA, USA) and Western blotting (Qiagen, Hilden, Germany) using anti-His antibodies.

### 2.4. Lipid Extraction from Bacteria

Extraction of total lipids from bacteria *P. gingivalis*, *T. forsythia*, *A. actinomycetemcomitans*, and *E. coli* was performed according to the protocol of Bligh and Dyer [59] with modifications described in Garbus et al. [60]. The pelleted bacterial cells were washed three times with 0.9% NaCl. To 200–400 mg of the pellet, 1 mL of dH_2_O and 4 mL of chloroform/methanol (1/2, *v*/*v*) were added. The mixture was vortexed and left to sit for 2 h at 25 °C. Chloroform (1.5 mL) and a mixture composed of 2 M KCl and 0.5 M K_2_HPO_4_ (1.5 mL) were then added and vortexed. The solution was centrifuged at 2000× *g* for 10 min to induce the separation of the upper aqueous phase and the lower lipophilic phase. The lower phase was left to evaporate, and the dried lipids were then dissolved in hexane/isopropanol/dH_2_O (6/8/0.75, *v*/*v*/*v*), vortexed, and centrifuged at 2000× *g* for 10 min. The upper phase was separated from the pelleted protein impurities, dried under nitrogen, and stored at −20 °C. Before use, the remaining lipids were dissolved in chloroform/methanol (9/1, *v*/*v*) and stored at −20 °C.

### 2.5. Lipidomic Analysis of a Total Extract of Lipids from Porphyromonas Gingivalis and Tannerella Forsythia

#### 2.5.1. Nuclear Magnetic Resonance Analysis

^1^H-NMR (400 MHz) spectra of *P. gingivalis* lipid extract were recorded in MeOH-d_4_ at 300 K on a nuclear magnetic resonance (NMR) spectrometer (400 MHz; Bruker, MA, USA), with a 5-mm double resonance broadband probe equipped with a pulsed-gradient field utility. The ^1^H-90° proton pulse length was 9.3 µs, with a transmission power of 0 db. The probe temperature was maintained at 300.0 K (±0.1 K) using a variable temperature unit (B-VT 1000; Bruker, MA, USA). Calibration of the chemical shift scale (δ) for ^1^H and ^13^C nuclei was obtained on the residual proton signal of the MeOH-d_4_ at δ_H_ 3.310 and δ_C_ 49.00 ppm, respectively. We obtained ^1^H-NMR (i.e., proton chemical shifts, scalar couplings). The resulting ^1^D and ^2^D-NMR spectra were analysed using TopSpin 3.6.1 (Bruker, Billerica, Germany). The lipid classes from the NMR data were identified through comparisons with our previous NMR measurements carried on commercially available lipid standards.

The details of the transitions used in the multiple reaction monitoring (MRM) experiments are provided in Appendix A.

#### 2.5.2. HPLC-Electrospray Ionization-Mass Spectrometry Analysis

The lipid extract was analysed by liquid chromatography-mass spectrometry (LC-MS) (Model 1100 series; Hewlett-Packard, Palo Alto, CA, USA) coupled to a quadrupole ion-trap mass spectrometer (Esquire LCTM; Bruker, USA, MA) equipped with an electrospray ionization source and in both positive and negative ion modes. Chromatographic separation of lipids was carried out at 303 K on a thermostated C_18_ column (Kinetex 2.6 µ; length, 100 mm; particle size, 2.6 µm; internal diameter, 2.1 mm; pore size, 100 Å; Phenomenex, Torrance, CA, USA). The solvent system consisted of eluent A as MeOH/H_2_O (7:3, *v*/*v*) containing 10 mM ammonium acetate and eluent B as isopropanol/MeOH (10:90, *v*/*v*) containing 10 mM ammonium acetate. Samples were resuspended in 1 mL CHCl_3_/MeOH (2:1, *v*/*v*), and 10 µL was run with a linear gradient from 65% eluent B to 100% B in 40 min, plus 20 min isocratic 100% B at 1 mL/min, to elute the diglycerides and triglycerides. The column was then equilibrated to 65% B for 10 min. The MS scan range was 13,000 U/s in the range of 50 to 1500 m/z, with a mass accuracy of ~100 ppm. The nebulizer gas was high purity nitrogen at a pressure of 20 to 30 psi, at a flow rate of 6 L/min, and at 300 °C. The electrospray ionization was operated in positive ion mode for the qualitative and quantitative analyses of phosphatidylcholine (PC), lyso-PC, and SM, and in both positive and negative ion modes for phosphatidylinositol (PI), phosphatidylethanolamine (PE), CPE, and DH-CPE. For the structural assignments of the lipid species, the extracted ion chromatograms from the positive and/or negative ion full scan data were integrated using the DataAnalysis 3.0 software (Bruker Daltonik, Billerica, Germany). Ceramide analyses were performed by multiple reaction monitoring (MRM) using a triple quadrupole mass spectrometer (SCIEX API 3000; Termo Fisher Scientific, Boston, MA, USA) coupled to an HPLC system (Shimatzu, Kyoto, Japan). Detailed parameters on transitions and collision energies are reported in the Appendix A. Column, eluents, and gradients were as described above. The calibration curve was calculated using the standard CPE (Matreya, PA, USA). Unfortunately, no commercial standards were available for 2′-acyl-dihydro-ceramide-phosphoethanolammine (2′-acyl-DH-CPE), so we could not obtain quantitative MS data for this lipid class by calibration curve. The exemplar LC-MS chromatograms of the lipid profiles from the untargeted lipidomics analyses are included in the Appendix A.

### 2.6. Clinical Periodontal Examination

Dental students (*n* = 20) aged 21–25 years with good oral hygiene and no gingival inflammation or systemic disease were recruited as the control group. For the periodontitis group (*n* = 20), patients with moderate periodontal disease (stage III, grade B) [61] were carefully selected. The inclusion criteria were: 25–70 years of age, untreated advanced periodontitis, a periodontal probing depth of ≥5 mm in at least four teeth in four different quadrants, and at least 20 teeth (excluding third molars). The diagnosis was supported by radiographic bone loss extending to at least the middle third of the root. Exclusion criteria were periodontal treatment in the past 12 months, use of systemic antibiotics, presence of fixed or removable prosthetic restorations or implants, pregnancy or lactation, or a systemic disease that might affect periodontal tissues, wound healing, or immune response (e.g., diabetes mellitus, bone metabolic diseases, HIV, immunosuppressive or anti-inflammatory therapy, or radiation). Smoking habits were recorded.

After a brief periodontal health and dental pathology interview at the Department of Oral Medicine and Periodontology, University Medical Centre Ljubljana, Slovenia, all patients underwent a comprehensive dental and periodontal examination performed by a single experienced examiner. Using the Williams periodontal probe (POW6, Hu-Friedy, Chicago, IL, USA), a calibration exercise for recession and probing pocket depth, including ten (*n* = 10) stage III periodontitis patients, yielded >95% of measurements within 1 mm of error. Periodontal parameters were recorded at six sites on each tooth using the same probe. These included the following parameters: absence/presence of plaque on tooth surfaces using a dichotomous plaque index, and absence/presence of bleeding on probing, probing pocket depth, and exercise for recession. Clinical attachment level was calculated *post hoc* from probing pocket depth and exercise for recession. If a peculiar non-periodontal reason for periodontal tissue destruction was suspected (e.g., endo-perio lesions, iatrogenic causes, orthodontic anomalies affecting the third molar distal to the second molar, gingival recession of traumatic origin, and dental caries in the cervical area), patients were excluded from the final evaluation.

The research proposal was approved by The National Ethics Committee (document KME 0120-142/2018/4). Furthermore, the principles outlined in the revised Declaration of Helsinki on experiments with human participants were followed.

### 2.7. Dental Plaque and Saliva Sampling

All subjects received detailed verbal and written instructions regarding saliva and dental plaque sample collection. These included not coughing or clearing their throat into the collection tube and abstaining from eating, tooth brushing, using a mouthwash, drinking alcohol, taking drugs, or smoking for 1 h prior to saliva collection. The mouth was rinsed with tap water a few min before saliva collection. Unstimulated saliva was collected into a sterile 15-mL tube until at least 1 mL of saliva was obtained.

The same dentist collected all plaque samples. Before sampling, selected teeth with the deepest probing pocket depth in each yaw quadrant were isolated with cotton rolls. The plaque sample was harvested with a sterile Gracey curette with one vertical stroke on the root surface starting from the periodontal pocket/sulcus depth. The working end of the curette was transferred to the sterile container and the visible plaque was wiped off against the wall of the container. Samples were collected from all four sites in each quadrant. In case of contamination of the plaque samples by blood, the samples were discarded and new plaque samples were collected from the second deepest probing pocket of the quadrant.

### 2.8. Total Lipid Extraction from Clinical Supragingival Plaque and Saliva Samples

Clinical samples of supragingival plaques (20 samples from healthy individuals and 20 samples from patients with periodontal disease) were first resuspended in 1 mL of dH_2_O and sonicated (Vibra-cell VCX 750, Sonics, USA, OK) on ice for 5 min at 40% amplitude under 2 sec cycles. Total lipids from clinical samples were extracted as previously described for bacteria (Section 2.4).

### 2.9. Thin-Layer Chromatography

Commercial lipids (SM, CPE) and total lipids extracted from *P. gingivalis*, *A. actinomycetecomitans,* and *E. coli* (10 µg each sample) were applied on a plate precoated with Silica Gel 60 (HPTLC plate). The plate was dried with a hairdryer and lipids were separated with a mobile phase consisting of chloroform/methanol/25% ammonia (65/25/4 *v*/*v*/*v*). After separation, the plate was dried again and the lipids were blotted to a polyvinylidene difluoride (PVDF) membrane using an iron. Previously, the PVDF membrane was activated for 30 sec in an activation solution (106 mg CaCl_2_ × 2H_2_0, 40 mL H_2_0, 14 mL methanol, 80 mL 2-propanol). After blotting the PVDF membrane was immersed in 3% bovine serum albumin (BSA) in Tris-buffered saline (TBS; 10 mM Tris-HCl, 150 mM NaCl, pH 7.5) for 2 h at 25 °C to block nonspecific binding of the protein. ErylysinA (2 μg/mL) in 3% BSA in TBS was added to the membrane and left to incubate for 2 h at 25 °C. The membrane was washed three times for 10 min in TBS and incubated with anti-His primary antibodies (1:1000) for 1 h at 25 °C. After washing, horseradish peroxide-conjugated anti-mouse IgG antibodies (1:1000) were added and left for 1 h at 25 °C. The membrane was washed three times for 10 min in TBS and the peroxide detection reagent was added. When the colour of the reaction product appeared, the surface of the membrane was washed with water.

To visualize the separated lipids, the HPTLC plate was sprayed with primulin reagent and the bands were visualized under UV light at 365 nm.

### 2.10. Dot Blot

The nitrocellulose membrane (Bio-Rad, USA) was cut into pieces (1 cm^2^ in size) and placed in a 12-well plate. Lipid extracts from bacteria (20 µg), clinical samples (200 µg), and lipid standards (14 µg CPE and 20 µg SM) were applied directly on individual nitrocellulose membranes and left for 15 min at 25 °C to dry completely. Each well was blotted with blocking buffer (3% milk in TBS) overnight at 4 °C with shaking (200 rpm). The plate was rinsed three times with TBS and incubated with EryA-mCherry (2 µg/mL, in 3% milk in TBS). After 2 h of incubation with shaking (200 rpm) in the dark at 25 °C, the plate was washed three times with TBS. Membranes were placed on glass slides, and the interaction of lipid-bound EryA-mCherry was analysed using a stereomicroscope (Leica MZ FLIII, Berlin, Germany) with a CCD camera at 1.0 × magnification. Images were analysed using the programme FIJI ImageJ. To quantify the binding of EryA-mCherry to lipid extracts from dental plaque, we measured the sum of all pixel intensities across all pixels in an object using ImageJ and an equimolar SM/Chol lipid mixture as a negative control to determine CPE-negative (healthy individuals) or CPE-positive individuals (periodontal patients).

The results of the dot blot assay from clinical plaque samples were analysed in relation to standard clinical diagnostic procedures to compare the results of both approaches using MedCalc statistical software (Ostend, Belgium).

### 2.11. Permeabilization of Small Unilamellar Vesicles

Small unilamellar vesicles composed of CPE, POPC, and Chol (molar ratio, 5/47.5/47.5) or of total lipids extracted from *P. gingivalis* or *E. coli*, were loaded with calcein in the self-quenching concentration (80 mM), as described previously [62]. The permeabilization was assayed using a fluorescence microplate reader Infinite F NANO (Tekan, Switzerland) with excitation and emission set at 485 and 535 nm, respectively. Vesicles were exposed to the protein mixture EryA/PlyB (EryA concentration 50 µg/mL; EryA/PlyB molar ratio 12.5/1). The rate of calcein release was measured for 30 min at 25 °C. The maximal calcein release was obtained after solubilisation of vesicles with 1 mM Triton X-100 and for each measurement, the percentage of calcein release was calculated.

### 2.12. Statistical Analysis

The results of the dot blot assay of clinical plaque samples were analysed concerning the standard clinical diagnostic procedures to compare both approaches by MedCalc (Version 20.110) statistical software (Ostend, Belgium). Standard diagnostic parameters (*sensitivity*: the probability that a test result will be positive when the disease is present (true positive rate), *specificity* (probability that a test result will be negative when the disease is not present (true negative rate)), *positive likelihood value* (ratio between the probability of a positive test result given the presence of the disease and the probability of a positive test result given the absence of the disease), *negative likelihood value* (ratio between the probability of a negative test result given the presence of the disease and the probability of a negative test result given the absence of the disease), *positive prediction value* (probability that the disease is present when the test is positive), *negative predictive value* (probability that the disease is not present when the test is negative), and *accuracy* (overall probability that a patient is correctly classified)) were calculated with a prevalence set at 50% (*n* = 20 in both groups).

## 3. Results and Discussion

Nowadays, it is challenging to diagnose the early stage of periodontal disease, and the search for a cost-efficient and effective agent for treating periodontal disease continues [11]. There is a great need for a periodontal disease-associated biomarker that would allow its early detection, however, no robust laboratory test has been developed so far [5]. One of the potential biomarkers for periodontal disease could be CPE, the major sphingolipid in the cell membranes of invertebrates, parasites, and some Gram-negative bacteria [63,64,65,66,67] and one of the most dominant sphingolipids in periodontopathogen species in phylum Bacteroidetes [68]. In vertebrate cell membranes, CPE is present only in trace amounts and is therefore almost undetectable in healthy individuals [43].

Even more, CPE species have also been suggested to be virulence factors of periodopathobiontic bacteria. These lipids mediate cellular effects via Toll-like receptor signalling, leading to secretion of interleukin 6 from dendritic cells and inhibition of osteoblast differentiation. This in turn leads to loss of teeth supportive bone, causing teeth to fall out. CPE species of bacterial origin have also been shown to promote other autoimmune diseases and have been found in numerous human tissues [44]. The importance of CPE species of bacterial origin leading to periodontal and other autoimmune diseases clearly demonstrates the importance of developing an assay that could measure the amount of CPE in biological human samples. Such an assay would allow the use of CPE as a prognostic marker for various CPE-associated inflammatory diseases.

### 3.1. Lipidomic Analysis of the Total Extract of Lipids from Porphyromonas Gingivalis and Tannerella Forsythia Show the Presence of CPE Species

*Porphyromonas gingivalis* was observed by ^1^H-NMR to have a major contribution of free ceramides (Cer, no phosphate head) and total Chol to its lipid composition, with a relative abundance of 29% and 71%, respectively (Figure 1). We could assess by MS the chain composition of the most abundant free Cer, such as Cer 42:2 (d18:1; 24:1, ca 22 mol% of total Cer); Cer 34:1 (d18:1; 16:0, ca 9 mol%); Cer 42:3 (d18:1; 24:2, ca 9 mol%); Cer 42:1 (d18:1; 24:0, ca 6 mol%); Cer 36:1 (d18:1; 16:0, ca 5 mol%); and Cer 41:2 (d18:1; 23:1, ca 4 mol%). Some Cer were found unsaturated and branched on the amidic chain, in accordance with the literature [50,52,69]. Total Chol was observed by MS as membrane-bound free Chol (FC) and as Chol-esters (CE). Free Chol accounted for only 8.0% of all Chol, while CE for the remaining 92%, with the major species CE 18:2 (ca 47 mol%); CE 18:2 (ca 20 mol%); and CE 18:3 (ca 12 mol%). The PC plus SM diagnostic ^1^H-NMR signal (3.21 ppm, singlet) was almost absent (<1%), PE diagnostic peak (3.10 ppm, multiplet) showed their relative abundance to be around 3%. Phosphoethanolamine was found by mass spectrometry to be composed only of branched, fully saturated acyl chains, with the major species PE 30:0 (ca 90%) and PE 28:0 (ca 10%).

CPE could not be observed, either by ^1^H-NMR or by LC-MS, due to their low abundance. On the other hand, mass analysis revealed the presence of 2′OH-CPE analogues (ca 2% of total lipids), with the three major species being 2′OH-DH-CPE 35:0 (ca 46 mol%); 2′-hydroxy-CPE (2′OH-CPE) 35:1 (ca 6 mol%); and 2′OH-DH-CPE 36:0 (ca 48 mol%) (Figure 2). The presence of hydroxylated CPE species and the absence of CPE in the lipidome of *P. gingivalis* was not surprising as quite similar results were already obtained in other studies [44,46].

Due to the small amount of lipid extract obtained from *T. forsythia* no NMR spectra could be acquired, meaning that inter-class relative abundance of lipid composition could not be obtained. We were able to achieve the lipid intra-class distribution by quantitative HPLC-MS/MS analysis to give a precise estimation of ceramide class abundance over total lipid extract. We focused on MS analysis of CPE analogues, and we could identify four CPE-related classes, bearing different oxidation states at the sphingosine double bond and at the 2′-position on the amidic chain. In particular, we observed canonical CPE; DH-CPE lacking the double bond (position C4-C5) on the sphingosine backbone; 2′-hydroxy-dihydro-CPE (2′OH-DH-CPE); and the 2′-acyl-dihydro-CPE (2′acyl-DH-CPE), that presented a branched acyl chain (Figure 3).

To estimate the concentration of CPE, DH-CPE, and 2′OH-DH-CPE, we assumed that their ionization response factor was the same, as determined by a calibration curve with a standard CPE. Unfortunately, the 2′-acyl-DH-CPE class was observed with a different MS setup, so we could not compare the MS data with the standard CPE. Quantitative MS analysis showed relatively low amounts of CPE and DH-CPE and a significant contribution of 2′OH-DH-CPE (Table 1). The concentrations of CPE and its analogues (% (*w*/*w*)) were calculated as the total mass of the lipid class over the total mass of the lipid extract. To obtain a qualitative estimate of the 2′-acyl-DH-CPE, we calculated the ratio of the total MS area of the 2′OH-DH-CPE species to the total MS area of the 2′-acyl-DH-CPE. Note that this was conducted assuming that 2′-acyl- and 2′-hydroxy-DH-CPE have the same ionization response. This gave a ratio of 2′OH-DH-CPE /2′-acyl-DH-CPE of 1.6, indicating that the acyl-DH-CPE concentration is at least of the same order of magnitude as the hydroxy-DH-CPE.

The most representative species of 2′OH-DH-CPE were found to be: 2′OH-DH-CPE 34:0 (% molar fraction = ca 37%); and 2′OH-DH-CPE 35:0 (ca 44%). A complete list of the most abundant Cer with their relative molar distribution can be found in Table 2. CPE and DH-CPE were much rarer than the hydroxylated analogues (total 0.04% *w*/*w*), with CPE 34:1 (ca 80% of all CPE) and DH-CPE 36:0 (ca 91% of all DH-CPE) accounting for the majority (Table 2). Interestingly, in addition to the masses of 2′OH-DH-CPE, we detected a series of ions 242 Da heavier than the corresponding hydroxylated species, suggesting the presence of 2′-esterified species by a C15 branched and saturated acyl chain (ω-2 methyl C14). We refer to this ceramide class as 2′-acyl-DH-CPE and, although we could not estimate their absolute concentration, we determined their intra-class relative distribution, finding two major species: 2′-acyl-DH-CPE 50:0 (ca 42%); and DH-CPE-ester 49:0 (ca 44%). It is worth mentioning the presence of odd chains in all Cer species, possibly due to ω-2 methyl branching of the amidic or 2′-acyl chains [44,46].

Besides DH-CPE, we observed other classes of dihydroceramides (DH-C) bearing different polar heads, such as DHC-phosphoglycerol, and DHC-phosphoinositol, as minor membrane components. In these classes, we detected the presence of the same C15 acyl chain at the 2′-position on the amidic chain. Interestingly, the relative distribution of DH-C and their corresponding C15 DH-C-esters followed a similar lipid species distribution, suggesting the acylation of the secondary –OH group as a biochemical process shared by all the 2′-hydroxyl-dihydroceramide classes. Regarding other lipid classes, we found total Chol as both FC (ca 7%) and CE (ca 93%), but we could not establish the lipid/Chol ratio as we had no NMR data available. The CE class was constituted by the major components CE 18:1 (ca 35%); CE 18:2 (ca 29%); and CE 18:3 (ca 20%). Very few species of PC and SM could be detected, suggesting a strongly favoured biosynthesis towards Cer with phosphoethanolamine head, rather than ceramide with phosphocholine head (i.e., SM). MS analysis of the free fatty acids (FA) class showed the most abundant species to be FA 18:0 (50%), FA 15:0 (23%), FA 18:1 (11%), and FA 16:0 (9%). It is interesting to notice that the second most abundant FA is the saturated ω-2 methyl C15 acyl chain, which is used upon 2-′acyl-DH-CPE biosynthesis.

The role and biophysical properties of dihydrosphingolipids are still enigmatic. What is certain is that in phospholipid–sphingolipid–cholesterol mixtures, DH-C gives rise to more rigid mixtures than the corresponding 4-unsaturated compounds (Cer) [70]. This enhanced rigidifying effect of DH-C was recently confirmed by other researchers [71] through combined molecular and cellular investigations; the same authors also demonstrated that DH-C were more potent in increasing membrane permeability than the unsaturated analogues. Another interesting, well-established molecular feature of DH-C is that, unlike the 4-unsaturated Cer, DH-C does not induce a transbilayer (flip-flop) lipid motion in membranes [72].

The presence of a hydroxyl group at position 2 of the fatty acid is a characteristic feature of the CPE species produced by both bacteria studied. However, the role of such modification of a fatty acid is far from clear and contradictory explanations are found in the available literature. Hydroxylation at position 2 of a fatty acid, associated with a sphingoid base has been implicated in *Saccharomyces cerevisiae* as a critical factor in maintaining membrane fluidity and proper turnover of membrane molecules [73]. *Porphyromonas gingivalis* and *T. forsythia* could utilize the modified CPE species to maintain membrane structure and function. On the other hand, other studies report that α-hydroxylation of the N-acyl chain has no significant effects on membrane order, although some relevant effects on the physical properties of pure Cer were observed when these hydroxylated sphingolipids were mixed with certain phospholipid mixtures [74].

As a final consideration, how it has been correctly emphasized in a recent review on the topic [75], we cannot avoid underlining the absence, in the majority of published papers on Cer, of the relationships between the pathophysiological observations and the physicochemical data. The first approach usually ignores the biophysical work, whereas the second approach (biophysical studies) does not give due attention to the physiological/pathological findings. This situation should be overcome soon if we want to better understand the molecular recognition phenomena in which these sphingolipids are involved [70,71,72,74,75].

### 3.2. EryA Specifically Interacts with Porphyromonas Gingivalis-Produced CPE Species

As lipidomic analyses showed the presence of CPE species in the total lipid extracts from *P. gingivalis* and *T. forsythia*, we wanted to further analyse whether commercial CPE and bacterially derived CPE species could be detected with EryA. We had difficulty culturing *T. forsythia*, which is known for its slow and fastidious growth [76], so we did not include lipid extracts from this bacterial species in further EryA binding studies.

The specificity of EryA interaction with CPE species was first analysed using thin-layer chromatography, followed by transfer of lipids to a PVDF membrane and study of EryA-lipid binding by immunodetection (Figure 4). Lipids from all bacterial species examined were successfully isolated and separated (Figure 4A). Immunodetection showed strong binding of EryA to commercial CPE and to the total lipid extract from *P. gingivalis*, whereas no binding was detected for lipids from *E. coli* and *A. actinomycetemcomitans* (Figure 4B). These results are in accordance with literature data, which reveal that the production of Cer is restricted to only a few bacterial species belonging to the phyla Bacteroidota (including *P. gingivalis* and *T. forsythia*), Fusobacteriota, Bdellovibrionota, Myxococcota, Mycoplasmota, and to the class of Alphaproteobacteria of the phyllum Pseudomonadota [77]. *Escherichia coli* and *A. actinomycetemcomitans* belong to the class of Gammaproteobacteria of the phyllum Pseudomonadota, for which there are no reports of ceramide synthesis. The weaker immunodetection signal of EryA bound to the CPE of *P. gingivalis* compared with the signal of EryA bound to commercial CPE is a consequence of the relatively low amounts of CPE species in the lipidome of *P. gingivalis* (approximately 2% of total lipids).

The specific interaction between EryA and CPE species was further confirmed by a dot blot assay in which EryA-mCherry was applied to commercial sphingolipids, CPE, and SM, as well as to total lipid extracts from *P. gingivalis* and *E. coli*. Binding of EryA-mCherry was visible on both CPE-containing samples (commercial CPE and *P. gingivalis* total lipid extract) but was not observed in the case of SM or *E. coli* (Figure 5). As in the case of thin-layer chromatography, the relatively weaker binding of EryA-mCherry to the lipid extract from *P. gingivalis* compared with commercial CPE can be explained by the small amounts of CPE produced by *P. gingivalis*.

It was previously shown that the combination of CPE-sensing EryA and its MACPF domain-containing protein partner PlyB permeabilizes artificial and natural CPE-containing membranes eventually leading to the lysis of CPE-containing cells [53]. This protein combination proved to be lytic to membranes containing less than 5 mol% CPE and showed selective toxicity to CPE-containing pests, including western corn rootworm and Colorado potato beetle larvae CPE, which contain only a few mol% CPE [53,58].

In this study, the membrane-permeabilizing potential of EryA/PlyB was investigated on vesicles prepared from the total lipid extract from *P. gingivalis* and *E. coli*, as well as on artificial vesicles with or without CPE (Figure 6). The highest permeabilization was observed in vesicles composed of commercial lipids and containing 5 mol% CPE, confirming the results of previous studies [53]. The EryA/PlyB protein mixture also permeabilized vesicles composed of the total lipid extract from *P. gingivalis*, although the permeabilization rate after 30 min was about 3-fold lower than that of artificial vesicles containing 5 mol% CPE. The much lower permeabilization effect of EryA/PlyB can be explained by the relatively small amounts of CPE species, which accounted for only 2% of the total lipids in *P. gingivalis*. The combination of EryA and PlyB showed no lytic activity on commercial vesicles lacking CPE (SM/Chol 1/1, mol:mol) or on vesicles composed of *E. coli* lipids, again confirming the specific activity of EryA for CPE and the absence of CPE in the *E. coli* lipid sample.

The combined results obtained by immunoblotting on separated lipids, dot blot, and calcein release assay strongly suggest that the affinity of EryA for lipid vesicles composed of *P. gingivalis* lipids is a consequence of the presence of CPE species, which were also identified in lipidomic studies. In addition, EryA was shown not to bind to negative controls containing SM or to lipids from bacterial species other than Bacterioidetes. This suggests that in the case of CPE detection by EryA in biological samples, contamination with blood or vertebrate cells would not affect detection.

### 3.3. EryA as a Sensor of CPE in Lipid Extracts from Plaque Samples from Periodontal Patients

Using the lipidomic approach, we confirmed the presence of CPE species in *P. gingivalis* and a specific interaction between EryA and CPE. Since periodontal disease in Slovenian patients is usually associated with the presence of *P. gingivalis* [78], we also wanted to test the possibility of detecting CPE species in clinical samples from periodontal patients using fluorescently-tagged EryA. Thus, the aim of this study was to evaluate the possibility of using EryA to detect CPE in clinical saliva or dental plaque samples from patients with periodontal disease; consequently, we could distinguish dental samples from healthy individuals from those with periodontal disease.

To test this hypothesis, 20 dental plaque and saliva samples were collected from healthy individuals and 20 from periodontal patients, and the lipids were isolated. Unfortunately, the amount of lipids isolated from the saliva samples was too low for adequate analysis, so we performed detection of CPE with EryA only for the plaque samples. As the amount of these biological samples was small, the semi-quantitative dot blot assay was used to determine the specificity of EryA for CPE for all samples in a single experiment. We observed the binding of EryA-mCherry to almost all (19/20, 95%) plaque samples from patients with periodontitis and to 11/20 (55%) plaque samples from healthy individuals (Figure 7). We assume that the presence of higher amounts of CPE in plaque samples from periodontal patients correlates with increased numbers of CPE-producing periodontal pathobionts (*P. gingivalis*, *T. forsythia*, *P. intermedia*) in dysbiotic biofilms belonging to the Bacteroidetes [10]. There was only one sample from the individual diagnosed with periodontal disease that was negative for EryA-mCherry binding. It may be possible that this negative patient plaque sample is the result of localised aggressive periodontitis caused predominantly by *A. actinomycetemcomitans* which has not yet been identified as a CPE-producing bacterial species [44,77,79]. Since periodontal bacteria can also be present in healthy individuals, albeit at lower amounts [80,81], some positive plaque samples from healthy individuals were expected.

The statistical analysis clearly shows that a negative dot blot assay confirms the absence of periodontitis. In contrast, a positive dot blot assay means that there is a 50% possibility one indeed has a periodontal disease.

Compared to the gold standard clinical diagnostic procedure, this results in a sensitivity of 95%, a specificity of 45%, a positive predictive value of 63%, and a negative predictive value of 90%, giving overall test accuracy of 70% (see Table 3 for details).

## 4. Conclusions

The results of our study show that *P. gingivalis* is indeed a CPE-producing bacterial species that can be specifically detected by CPE-sensing EryA. Moreover, CPE-sensing EryA is able to detect CPE not only in commercial CPE-containing vesicles or in lipid extracts from periodontal CPE-producing bacteria *P. gingivalis*, but also in biological samples presumably containing CPE-producing bacterial species. We show here that aegerolysin EryA can be used in combination with the dot blot technique as a suitable semi-quantitative method to determine the presence of CPE in clinical supragingival plaque samples and to detect periodontal disease. However, due to the low amount of CPE in clinical samples, it is difficult to accurately predict whether someone actually has periodontal disease or not. Our research shows the potential of non-self-lipids as new molecular biomarkers for bacterial-associated diseases. Even more, EryA, in combination with its protein partner PlyB, forms pores in vesicles composed of total lipid extracts from *P. gingivalis*, suggesting that the protein complex may have antimicrobial activity. Finally, because diagnostic methods for the determination and quantification of non-self lipids, which are present in relatively low amounts, are not yet well established, we propose the further development of lipid-based sensing techniques that would allow a more accurate and sensitive analysis of these non-self lipids as biomarkers for chronic inflammatory diseases. Lipid sensing proteins, especially proteins from the aegerolysin family, therefore have great potential for use as diagnostic tools or biosensors in biomedicine.

## Figures and Tables

**Figure 1 membranes-12-00655-f001:**
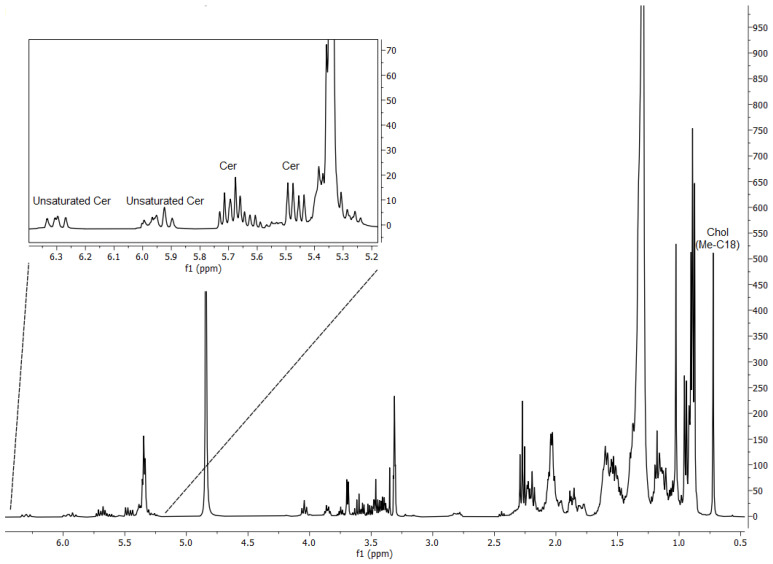
^1^H-NMR of *Porphyromonas gingivalis* raw lipid extract. Signals relative to ceramides (Cer) and total cholesterol (Chol) are highlighted.

**Figure 2 membranes-12-00655-f002:**
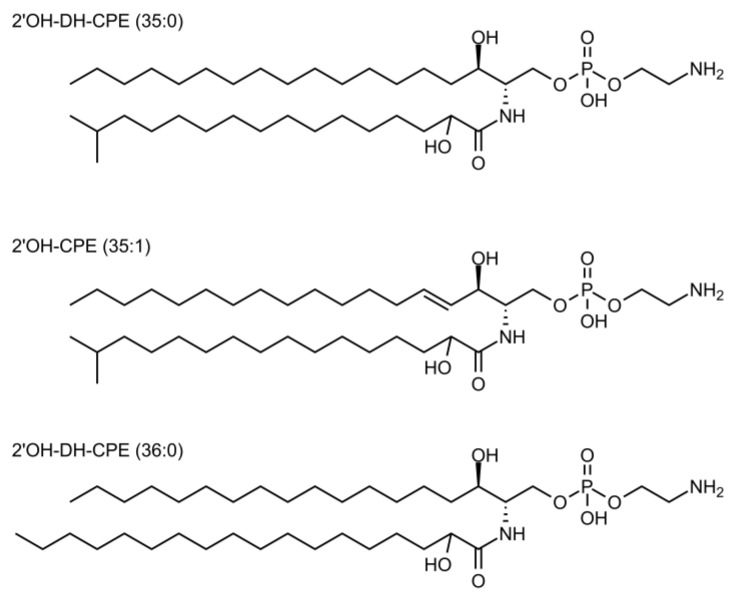
Structures of most representative *Porphyromonas gingivalis* CPE analogues.

**Figure 3 membranes-12-00655-f003:**
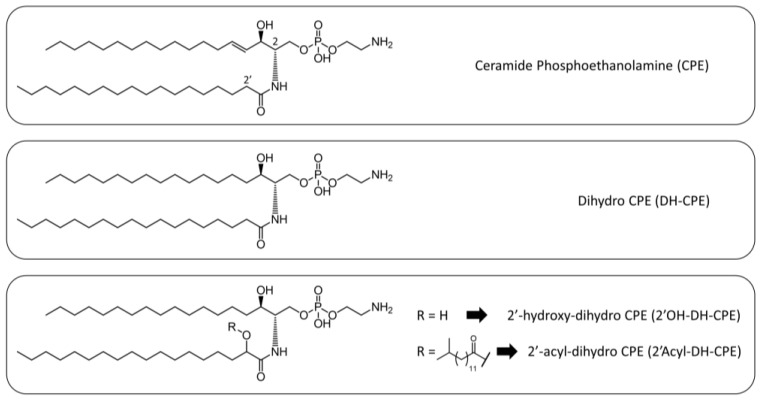
Structures of CPE analogues isolated from *Tannerella forsythia*. A CPE is shown at the top; a reduced dihydro CPE in the middle (DH-CPE); and a 2′-hydroxylated or 2′-acylated DH-CPE (2′OH- or 2′-acyl-DH-CPE) at the bottom.

**Figure 4 membranes-12-00655-f004:**
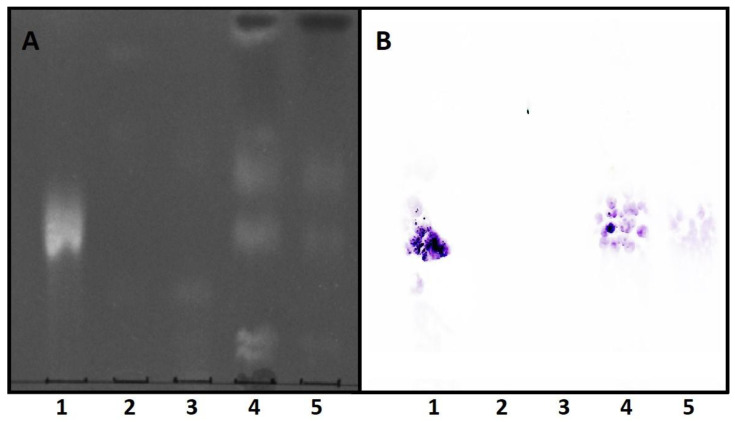
Thin-layer chromatography and immunodetection of CPE-bound EryA. Lipid standard CPE (1) and total lipid extracts from *E. coli* (2), *A. actinomycetecomitans* (3), *P. gingivalis*, nonpolar fraction (4), *and P. gingivalis*, aqueous fraction (5) were separated on an HPTLC plate with a solvent mixture of chloroform/methanol/25% ammonia (65/25/4 by volume). Primulin reagent was sprayed for detection of lipids under UV light (**A**). The blotted PVDF membrane was incubated with EryA (2 µg/mL) which was immuno-detected using peroxidase-conjugated secondary antibodies (**B**).

**Figure 5 membranes-12-00655-f005:**
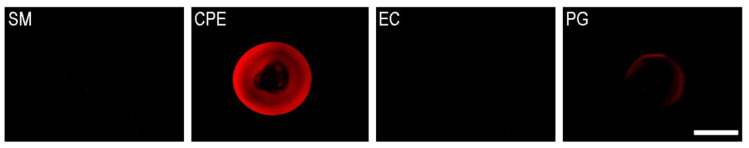
Dot blot images of EryA-mCherry binding to commercial lipids and lipid extracts from bacteria. Lipid standards (20 µg SM, 14 µg CPE) and 20 µg of lipid extracts from *E. coli* and *P. gingivalis* were applied on a nitrocellulose membrane, incubated with EryA-mCherry (2 µg/mL) and the intensity of fluorescence was measured. Scale bar: 2.5 mm. Abbreviations: SM, sphingomyelin; CPE, ceramide phosphoethanolamine; EC, *E. coli*; PG, *P. gingivalis*.

**Figure 6 membranes-12-00655-f006:**
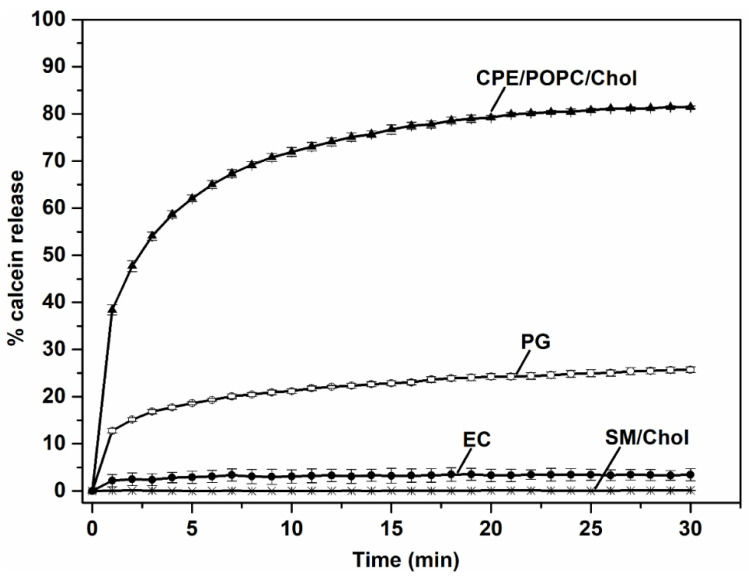
Calcein release from vesicles composed of different lipids. Permeabilization of small unilamellar vesicles composed of CPE, POPC, Chol (5/47.5/47.5, molar ratio) and SM, Chol (equimolar ratio) and of total lipid extract from *E. coli* and *P. gingivalis*. Fluorescence intensity of calcein released from the vesicles was monitored as described in Methods. Data are means ± SE from three independent experiments. Abbreviations: SM, sphingomyelin; CPE, ceramide phosphoethanolamine; POPC, 1-palmitoyl-2-oleoyl-*sn*-glycero-3-phosphocholine; Chol, cholesterol; EC, *E. coli*; PG, *P. gingivalis*.

**Figure 7 membranes-12-00655-f007:**
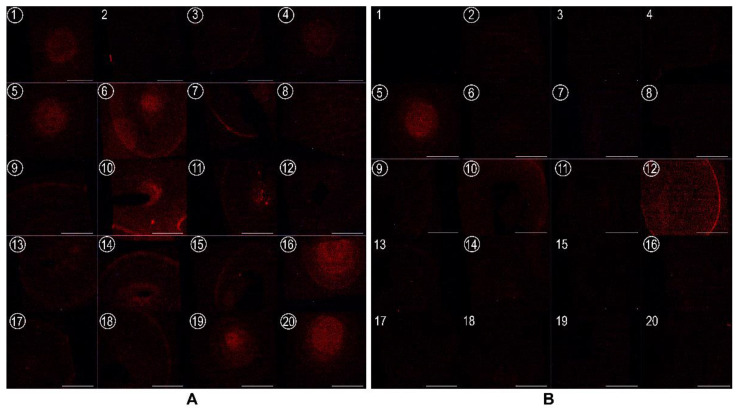
Dot blot images of EryA-mCherry binding to total lipid extracts from clinical supragingival plaque samples from patients and healthy individuals. Twenty samples (**A**) from patients with periodontal disease and twenty (**B**) from healthy individuals (200 µg lipids) were applied on a nitrocellulose membrane, incubated with EryA-mCherry (2 µg/mL) and the intensity of fluorescence was measured. Scale bar: 5 mm. Positive samples are indicated by a number within the circle.

**Table 1 membranes-12-00655-t001:** Concentrations (% *w*/*w*) of ceramide classes CPE, DH-CPE, and 2′OH-DHCPE of total lipid extract from *Tannerella forsythia*.

Class	Relative Inter-Class Distribution (% *w*/*w*)
CPE	0.02
DH-CPE	0.02
2′OH-DH-CPE	26.6
2′-acyl-DH-CPE	N/A

**Table 2 membranes-12-00655-t002:** Intra-class relative distribution of lipid species in the four ceramide classes from *Tannerella forsythia* (% molar fraction).

Species	Relative Intra-Class Distribution (% Molar Fraction)
CPE (32:1)	8.3
CPE (33:1)	11.2
CPE (34:1)	80.4
DH-CPE (34:0)	8.9
DH-CPE (36:0)	91.1
2′OH-DH-CPE (33:0)	8.6
2′OH-DH-CPE (34:0)	37.0
2′OH-DH-CPE (35:0)	43.6
2′OH-DH-CPE (36:0)	9.0
2′-acyl-DH-CPE (48:0)	9.9
2′-acyl-DH-CPE (49:0)	43.6
2′-acyl-DH-CPE (50:0)	41.7
2′-acyl-DH-CPE (51:0)	4.3

**Table 3 membranes-12-00655-t003:** The statistical relationship between the results of the dot blot assay of clinical plaque samples compared with clinical diagnosis.

Statistic	Value	95% Cl
Sensitivity (True positive/ True positive + False positive)	95.00%	75.13% to 99.87%
Specificity (True negative/True negative + False positive)	45.00%	23.06% to 68.47%
Positive likelihood ratio (Sensitivity/1 − Specificity)	1.73	1.15 to 2.6
Negative likelihood ratio (1 − Sensitivity/Specificity)	0.11	0.02 to 0.8
Positive predictive value (True positive/ True positive + False positive)	63.33%	53.43% to 72.22%
Negative predictive value (True negative/ True negative + False negtive)	90.00%	55.64% to 98.48%
Accuracy (True positive + True negative/ True positive + True negative + False positive + False negative)	70.00%	53.47% to 83.44%
Disease prevalence	50.00%	

## Data Availability

The data presented in this study are available on request from the corresponding author.

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
