# Peer review of "Ceramide Phosphoethanolamine as a Possible Marker of Periodontal Disease"

_membranes, 2022, doi:10.3390/membranes12070655_

Round 1
Reviewer 1 Report
Skočaj et. al reported work "Ceramide phosphoethanolamine as a possible marker of perio-dontal disease" interesting can be acceptable for the publication in Membranes after addressing minor comments as specified below.
1) Although introduction looks interesting, it has too much of garbled information. I rather recommend the authors shorten the introduction section by highlighting the significant content which is more relevant to current research.
2) There are some typographical mistakes, need to be corrected.
3) NMR spectra Figure. 1 inset not visualising properly, therefore high resolution figure need to present in the inset.
4) Figure 4 looks very crude, I rather recommend to redo the experiments and present proper picture. Current picture bands looks like diffused or contaminated.
Author Response
Reviewer #1:
Query #1:
Although introduction looks interesting, it has too much of garbled information. I rather recommend the authors shorten the introduction section by highlighting the significant content which is more relevant to current research.
Reply:
We agree with the reviewer #1, there were some information in the submitted version of the manuscript which are not very relevant for the study. We have significantly shorten the introduction and all the changes are marked with track changes.
Query #2:
There are some typographical mistakes, need to be corrected.
Reply:
Indeed there were some typographical mistakes. We have corrected the mistakes which are marked with track changes.
Query #3:
NMR spectra Figure. 1 inset not visualising properly, therefore high resolution figure need to present in the inset.
Reply:
We have prepared a new version of Figure 1.
Query #4:
Figure 4 looks very crude, I rather recommend to redo the experiments and present proper picture. Current picture bands looks like diffused or contaminated.
Reply:
Figure 4 represents a Far Eastren blot. To do this, the lipids are first resolved on TLC plate and then transferred (blotted) to PVDF membrane using an iron, preheated to 180 °C. At this temperature, the transferred lipids are “baked/dissolved/cooked” and this is the reason why the Figure 4 looks like being contaminated. We hope that the explanation satisfies the reviewer #1.
Reviewer 2 Report
The manuscript by Grundner et al describes experiments aimed at evaluating the utility of the aegerolysin protein, erylysin A (EryA) to detect ceramide phosphoethanolamine (CPE) species as a means of monitoring periodontal disease. A variety of experimental techniques were employed for the profiling of lipids from bacteria and plaque samples including nuclear magnetic resonance (NMR) spectroscopy and liquid chromatography-mass spectrometry (LC-MS). Data sets have been presented detailing that the periodontal pathogens, Porphyromonas gingivalis and Tannerella forsythia synthesize CPE. In addition, the authors have reported that the CPE-containing bacteria can be detected using EryA and that the protein was able to determine the presence of CPE species in clinical samples of dental plaque from periodontal patients. This is an interesting study although there are a few points that should be addressed.
1. Materials and Methods. A section on statistical analysis performed in the study should be included providing full details of the specific tests.
2. Materials and Methods. What was the purpose of extracting the samples with isopropanol and hexane following the initial isolation of lipids using the Bligh-Dyer extraction?
3. Materials and Methods. Details of the transitions used in the multiple reaction monitoring (MRM) experiments should be provided. This could be included in a supplemental section
4. Results and Discussion. It would be useful to include exemplar LC-MS chromatograms of the lipid profiles from the untargeted lipidomics analyses.
5. Results and Discussion. The term ‘canonical CPE’ and ‘canonical DH-CPE’ is not clear and should be further explained in the text.
Author Response
Reviewer #2:
Query #1:
Materials and Methods. A section on statistical analysis performed in the study should be included providing full details of the specific tests.
Reply:
A section on statistical analysis performed is now included in the text as a last chapter in the section Material and Methods. There are provided all the details of the analysis.
Query #2:
Materials and Methods. What was the purpose of extracting the samples with isopropanol and hexane following the initial isolation of lipids using the Bligh-Dyer extraction?
Reply:
The additional extraction gives a purer organic phase with less/without protein impurities as already explained in the text.
Query #3:
Materials and Methods. Details of the transitions used in the multiple reaction monitoring (MRM) experiments should be provided. This could be included in a supplemental section
Reply:
The details of the transitions used in the multiple reaction monitoring (MRM) experiments are now provided and are included into supplemental document (Table S1 and S2).
Query #4:
Results and Discussion. It would be useful to include exemplar LC-MS chromatograms of the lipid profiles from the untargeted lipidomics analyses.
Reply:
We have included the TIC chromatograms in the supplemental document as Figure S1.
Query #5:
Results and Discussion. The term ‘canonical CPE’ and ‘canonical DH-CPE’ is not clear and should be further explained in the text.
Reply:
We have removed the word "canonical" from the text, as it was indeed leading to confusion and it is not relevant.